# COVID-19 Vaccination: What Do We Expect for the Future? A Systematic Literature Review of Social Science Publications in the First Year of the Pandemic (2020–2021)

**Lorenzo Pratici** [1,]*[ID] **and Phillip McMinn Singer** [2]

1 Department of Economics and Management, University of Parma, 43125 Parma, Italy
2 Department of Political Science, University of Utah, Salt Lake City, UT 84112, USA; phillip.singer@poli-sci.utah.edu
* Correspondence: lorenzo.pratici@unipr.it; Tel.: +39-0521-032284 or +39-3484254139

**Abstract:** The Covid-19 pandemic has had wide-reaching societal and economic effects and a return to "normal" will take years to accomplish. In light of this situation, the most important advancement since COVID-19′s emergence has been the development of multiple, life-saving, vaccines. Academic research on vaccine has been extensive. It is estimated that in only one year it has been produced more published and indexed papers on this single issue than in the last twenty years on any other single issue, thus, necessitating some organization. This research consists of a systematic literature review of the social science publication on COVID-19 published in the first year of the pandemic (February 2020 to March 2021). This review is important because it occurs at a time when vaccines have begun their global distribution and the best efforts to address the pandemic is through vaccination programs. In this research, 53 papers published in relevant journals are analyzed out of the almost 30,000 articles retrieved from Scopus database. The analysis conducted relies on two different types: descriptive analysis (evolution at the time of citations; evolution over time of keywords; bibliographical mapping of countries, the top 10 most influential papers), and bibliometric analysis for content evaluation. A cluster analysis was performed for the latter. Clustering the research papers, based on the actual content of papers, found there to be five research areas: (1) economic aspects; (2) ethics and legal aspects; (3) health communication; (4) policies and crisis management, and (5) political issues. Yet, this article's results paint a picture of literature that has not yet considered the full scope of COVID-19′s effect on the economic, political, and population level health and well-being. Nor has it considered these effects across the global community, suggesting new potential areas of research and giving a perspective of what we should expect for the future.

**Keywords:** vaccine; Covid-19; Novel Coronavirus; social science; systematic literature review; PRISMA

## 1. Introduction

The COVID-19 pandemic caused by the spread of the SARS-CoV-2 virus, has had wide-reaching societal and economic effects. In the United States alone, the estimated cost of the pandemic is USD 16 trillion [1] and the number of unemployed in the country more than tripled [2]. The pandemic will cost the European Union almost EUR 1 billion, close to 6% of its gross domestic product [3]. As of May 2021, there have been more than 165 million cases and 3.4 million deaths [4]. Returning to "normal" will take years to accomplish, as economic and societal dynamics across the globe continue to be threatened by the pandemic.

Perhaps the most important advancement since COVID-19′s emergence, and the key to a return to a semblance of "normalcy", has been the development of multiple, life-saving, vaccines. As Dr. Tedros Adhanom, Director-General of the World Health Organization remarked, "[t]he development and approval of safe and effective vaccines less than a year

after the emergence of a new virus is a stunning scientific achievement, and a much-needed source of hope" [5].

Yet, the unprecedented speed in developing COVID-19 vaccines has introduced a series of new challenges for policymakers to consider. As Director-General Adhanom laments after highlighting the hope that comes from the development of vaccines, it is imperative for policymakers to address disparities in access to vaccines or risk further dividing the global community. The rapid development of a COVID vaccine has introduced many topics which social science researchers have only begun to contemplate and analyze. This includes the ethics of vaccine distribution, uptake, communication, and the financial and economic impacts of the development of the vaccine. While researchers have individually begun to research these topics, to date there has not been a systematic review conducted. Here we present the first systematic review of social science literature on COVID-19 vaccines. Our analysis will contextualize and understand the emergent literature, highlight strengths of the social science response to COVID-19 vaccinations, and identify areas for researchers to focus on in the future. This systematic review is particularly important because it occurs at a time when vaccines have begun their global distribution and the best efforts to address the pandemic is through vaccination programs.

Our review takes the following form. First, we provide a literature review on the background of COVID-19, systematic reviews, and vaccinations. Second, we provide an overview of our methodological approach. Third, we discuss the results of our review. Lastly, we highlight some of the implications and considerations in light of our analysis here.

## 2. Literature Background

Vaccines stand as one of the greatest public health interventions ever developed [6].

Because vaccinations have had such a monumental impact on public health, there has been a vast literature developed on the subject. For concision, we highlight three strands of interrelated literatures here, with a particular emphasis on systematic reviews that have previously been conducted on the topic of vaccinations.

First, COVID-19 has generated a massive amount of research. According to Google Scholar, nearly two hundred thousand articles have been published on the topic since 2020. This literature is expansive, covering a variety of different topics within the pandemic, including health effects, etiology, and comparisons of experiences and policies across different countries. However, because the volume of research produced is so expansive, conducting systematic reviews is essential to help understand trends in research and to evaluate research findings [7].

Second, prior to COVID-19, social scientists have conducted a vast set of systematic reviews on vaccinations. In particular, there has been published research on influenza [8–10], human papillomavirus [11–13], MMR [14–16], and H1N1 [17,18]. As highlighted above, these reviews have found that the development of vaccines and vaccination programs has had widespread and important improvements to public health and health care globally. The positive benefits of vaccines are consistent across disease type, though researchers have highlighted the potential for disparities by income and infrastructure within countries.

Third, amongst these well documented systematic reviews on vaccines, there are several key themes which have emerged. One of the most well developed research agendas leveraging systematic reviews has included vaccine hesitancy [10,19–21]. This work has largely found that there are few existing strategies to address hesitancy and little to quantify interventions. However, there is some evidence that leveraging dialogue-based interventions, such as social and mass media and non-financial incentives, have been successful at ameliorating hesitancy. Additionally, researchers have previously conducted systematic reviews on factors associated with vaccine uptake across a variety of different populations [22–28]. Across these reviews across different populations, researchers have found that the perceived effects of the vaccine, the likelihood of infection of an infectious disease, and the individual attitudes to vaccines are important in explaining uptake, and physician recommendations.

In light of this, a Systematic Literature Review (SLR) seems to be an appropriate start to organize what has been done so far. However, since the emergency has covered only a short period of time of just over a year and there were no previous studies before the beginning of such emergency, a clear strand focused on COVID-19 vaccination is social sciences has yet to be built. As such, analyzed papers will refer mainly to general issues caused by the pandemic. The discriminatory point on whether to include them in the sample or not is based on if and how they dedicate a part of their content to COVID-19 vaccination. Published papers that do not focus explicitly on COVID-19 vaccination but have reserved at least one paragraph to assess direct consequences of a vaccination campaign can therefore be included in the sample. However, we exclude manuscripts that merely mention COVID-19 while focusing on a different topic.

## 3. Method

This paper relies on SLR methodology to select 53 papers identified in relevant journals out of the almost 30,000 articles retrieved from Scopus database.

This particular method has been chosen for two main reasons. First, SLR methodology focuses on the transparency of the process and the replicability of the search, results, and analysis. This is particularly important in reducing biases and improving the methodological strength of the analysis [29,30]. Second, the recency of the topic of our analysis enables SLR to integrate and summarize the common elements of this developing body of research, as well as contrasting the differences that have emerged [29].

This is made necessary due to the fact that analyzing the social impact of COVID-19 vaccinations is, unfortunately, characterized by a limited timeframe. Yet, because of the exponential growth in research focused on COVID-19 and vaccinations [31] as well as the impact in terms of publications that this issue is having [32], it is a suitable topic for a literature review [31,32].

Additionally, our analysis supplies an important "snapshot" of the development of social science research during the first year and a half of the pandemic—with a particular focus on the impact of the vaccine, public opinion reactions, and the organizational processes supporting a global vaccination campaign. The literature examining COVID-19 and vaccinations will continue to evolve over the coming years and our work will provide an important marker for comparison to these future developments for policymakers to assess their actions and for health organizations planning to address COVID-19 in their communities.

### 3.1. Sampling

We analyzed researched published in Scopus, since it is the broadest social science database, as highlighted by Mishra et al. [33]. As Fanelli et al. [34] found, other databases such as Web of Science only index 12,000 peer-reviewed journals, while PubMed may exclude those papers that may be affiliated to political science, management or other social science journals. Scopus, instead, includes more than 20,000 peer-reviewed scientific journals, providing the broadest possible coverage [33,35].

The chosen dataset is wider than what is needed for the research; however, as papers are subject to several inclusion criteria defining them as relevant or non-relevant to the scope of the analysis. In other words, out of the whole sample identified on Scopus, we had to determine what articles should be included as part of the sample.

In order to identify the final sample, we used the PRISMA flow diagram method. This methodological approach was devised by Moher et al. and subsequently used by different authors [34,36]. The PRISMA flow diagram helps us to systematize and identify the final set of articles to work on following a 4-step approach: (1) identification of works, meaning the search of available records on identified database; (2) screening of suitable articles, meaning the act of excluding duplicates by reading titles and abstracts; (3) eligibility of publications, meaning the automatic exclusion of non-relevant paper, classified in other disciplines by the same Scopus database, as well as a further reading of full remaining

papers in order to classify them as suitable or non-suitable for the analysis; (4) inclusion of the articles that will be subject to the analysis, namely the final identification of the papers to be processed. The PRISMA flow used in this work is represented in Figure 1.

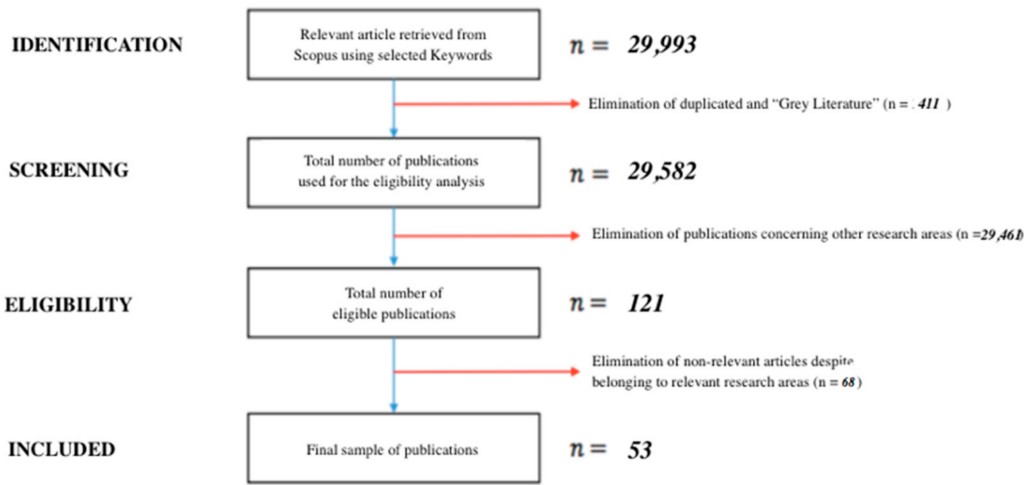

**Figure 1.** PRISMA Flow process applied for the research.

Authors retrieved articles related to the impact of COVID-19 vaccination with a Boolean research conducted on Scopus. The following keywords were used: *"COVID 19"* OR *COVID19\** OR *COVID-19\** OR *SARS-CoV-2\** OR *"Novel Coronavirus"* AND *Vaccination\** OR *Vaccine\**. The keywords we selected needed to be present in the title of the paper or in the abstract to be included in the results. We collected all articles that returned from these search criteria from the time of the outbreak of COVID-19 through 20 March 2021. This resulted in a full dataset of 29,993 indexed papers. From this full dataset, we excluded 411 articles in the grey literature (conference proceedings, book chapters, etc.). We then selected articles published in social science journals, basing our choice on the classification made by Scopus, which resulted in 121 articles. The remainder of the articles were published in medical and clinical journals.

The final step of the PRISMA model, consisting in the eligibility and assessment of papers, has been conducted by two Ph.D. trained researchers, which finalized the sample for the bibliometric analysis.

The rate of agreement between the assessment made by the two researchers has been calculated identifying the Cohen's Kappa coefficient, which measure the inter-rater reliability of choices [34]. To determine this index, we followed the methodology designed by Cohen in 1960. The two researcher-judges had to use a chromatic semantic-dimensional scale of 1 to 5 to indicate how relevant the article was (dark red = 1, non-relevant; orange = 2, very low relevancy; yellow = 3, possible relevancy; light green = 4, relevant; dark green = 5, very relevant). The judgement was double-blind. During comparison of the scoring of articles, any discrepancy of more than 2 points, the researchers conducted a discussion on the manuscript. Finally, the Cohen's Kappa was calculated to determine the inter-reliability of the choices and with a 0.8794 Cohen's K, this score has been considered acceptable. As shown in Figure 1, after this process, the final dataset includes 53 peer-reviewed papers, ranging from 2020 to 2021.

The short temporal dimension, which may represent a bias, is due to the recency of the topic. However, according to Brodeur et al. [32], the amount of scientific works concerning such a specific topic (the COVID-19 pandemic) is comparable to the amount of papers produced on any other specific topic in the last 25 years. No specific evidence is present on the vaccination topic addressed by this present work, and to the best of our knowledge this represents the first attempt of a literature review on the issue.

*3.2. Research Design*

The research is structured based on two different types of analysis. First, we conducted a descriptive analysis of the included articles. Second, we conducted a bibliometric analysis for content evaluation performed using VosViewer® software.

In the descriptive portion of the analysis, the results section reports the ranking of major articles by number of citations, the major topics emerging from the literature so far about the social impact of vaccination (this analysis is performed in a joint double-blind coding process), the ranking of countries per publication, and the ranking of journals per publication and the general trend of citations.

We descriptively assess the impact of time over citations, following the [37] methodology of usage of the citation per year (CPY) index. This index allows for taking into account the lag time for citations that biases older articles if compared to more recent publications [34].

This is necessary even though publications concern only a less than two-years timeframe. Indeed, if we consider the amount of publications on this issue, we can appreciate how the global number is comparable to another topic subject to a SLR. We modify the CPY to classify papers from a per-year rate to a per-month, constructing a citation per month (CPM) index, using the following formula.

Equation (1)—Determination of CPM Formula.

$$CPM = \frac{Cit}{(16 - mop)} \qquad (1)$$

where *Cit* stands for "number of citations" and *mop* stands for "month of publication". To calculate *CPM*, we divide the number of citations by how many months the article has been published. Since we took into consideration articles published starting from January 2020, and we currently are—as we write—in March 2021, this is the 15th month from the official beginning of the pandemic threat [38]. As such, the coefficient 16 represents the current month +1 from the pandemic beginning.

Additionally, we analyze article keywords so that it allows us "to analyse very large amounts of text without losing touch with focusing on small amounts of the material in considerable depth', according to Silverman [39]. Fanelli et al. [34] argues that "it also makes it possible to overcome the issue of selection bias, which is very frequent in literature reviews". This advantage can be very useful in an analysis based on a shorter time frame. The literature finds that bias based on a minimum amount of "false positive" (considered as sources identified but not relevant to the analysis) may be overcome by using this tool and it gives an added value to the research.

In general, practitioners search for literature based on topics or keywords as they are needed [39]. Clustering recurrent keywords, according to the number of co-occurrences, provides valuable information about where the literature has focused and what directions they are following.

The second type of analysis consisted of a bibliometric approach. In this portion of the analysis, we follow Christoffersen [40] in conducting a content evaluation of the included articles. In particular, we are interested in applying a visualization of similarities technique, developed by Van Eck and Waltman [41] and leveraged recently by Fanelli et al. [34]. Using VOSViewer software, we map the distance between two objects, in this case, the included papers, by similarity of the topic. Additionally, we analyzed the included papers through bibliographic coupling analysis, which calculates the overlapping literature cited by the papers in our sample. Lastly, members of the research team carefully read all the included papers and classified them into different research areas (RAs). This allowed as to determine what literature strands seem to be more developed at the present time and where a contribution is still needed for future literature.

## 4. Results

### 4.1. Descriptive Results

4.1.1. Evolution in Time of Published Works and Citations

The number of publications increased from the beginning of the emergency until December 2020, as shown in Figure 2a. Not surprising, we find that social science publications on COVID-19 have increased over time, with an exponential inflection between August and October 2020. This trend flattens between November and December 2020, reaching a plateau and then decreasing in the first quarter of 2021.

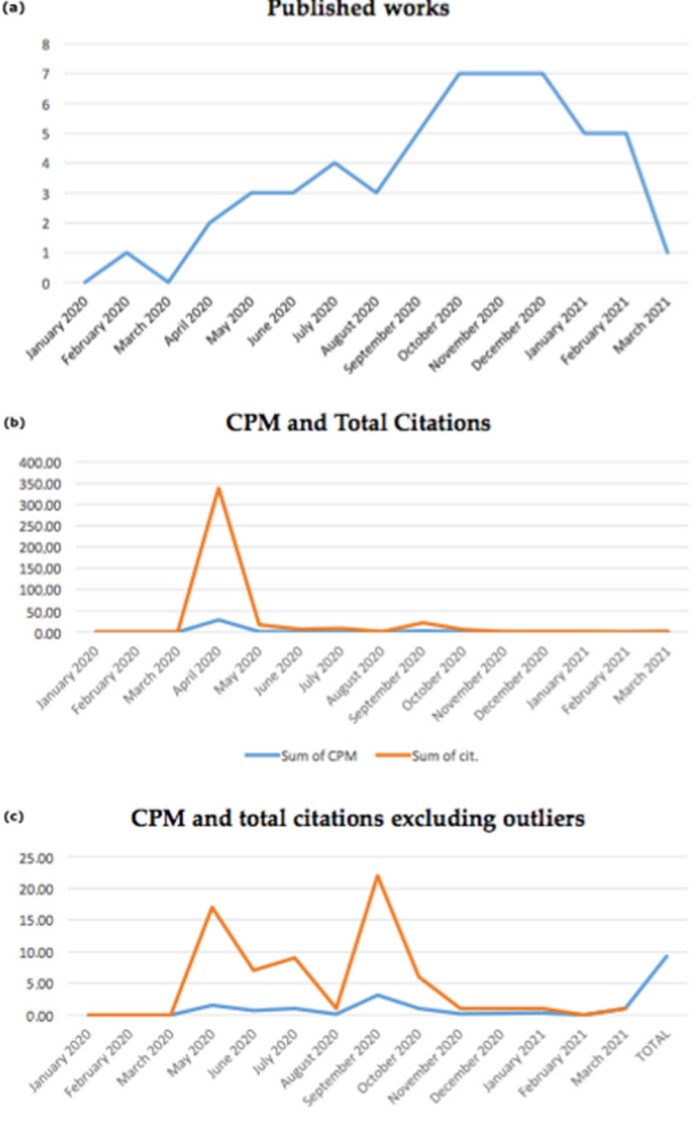

**Figure 2.** Number of published works and number of citations and weighted citations obtained by papers in the sample.

The trend for the evolution over time of citations is also represented by Figure 2b. Social science researchers demonstrated their rapid undertaking of COVID-19 and vaccines, beginning publication on the topic in February 2020. The number of citations alone, however, does not provide any information about the lag time speculated above [34,37,42] and figures also reported the CPM index trend, which substantially confirm, with less emphasis, the same trends identified for the total number of citations.

Citations and CPM have not increased, with a mean of citations corresponding to 30.58. However, the presence of an outlier makes the evaluation of CPM changes over time harder to be done. The graph clearly shows a significant difference between the month of April 2020 and all other months (CPMsum-April2020 = 9.30 against an overall CPMsum of 37.46 and CPMmean-April2020 = 0.27 against an overall CPMmean-April2020 = 1.31). This indicates that the paper having had the main impact on CPM was published in April 2020, followed by September 2020 (with a CPM sum = 3.14 and a CPMmean = 0.63). To avoid this discrepancy, Figure 2c excluded the month of April, and the outlier paper, "Pandemics, tourism and global change: a rapid assessment of COVID-19" by Gossling et al. [43], which resulted in the new mean of citations without considering the latter consists of 2.38.

### 4.1.2. Evolution of Keywords

We also find several trends in recurrent keywords describing the social sciences literature related to COVID-19 and vaccines. Figure 3 conveys keyword co-occurrence and their evolution over time. The map shows the most used keywords, with the larger dots representing more frequently used keywords. Additionally, Figure 3 also represents the connections between each keyword. The closer that a keyword is to another keyword indicates a higher level of link-strength (LS) between the keywords. In the case of keywords occurrence, an LS indicates the number of publications in which the terms occur included in the original sample.

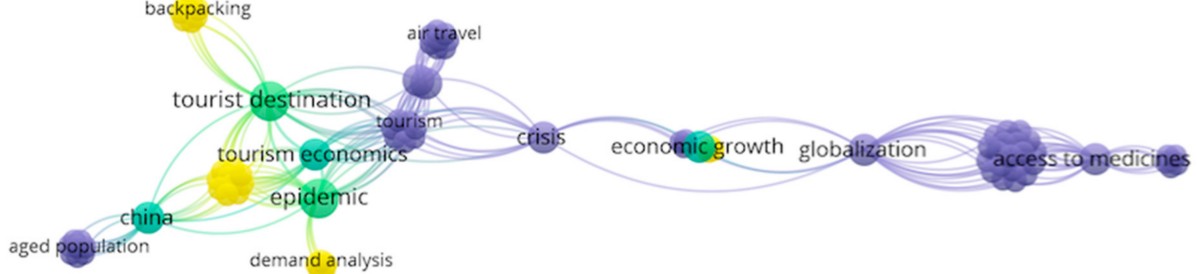

**Figure 3.** Keywords bibliographical map.

To determine the most cited keywords, all the words already used in the Boolean research to determine the sample have been excluded (namely: "COVID 19"; COVID19; COVID-19; SARS-CoV-2; Coronavirus*; "Novel Coronavirus"; Vaccination* and Vaccine*). Nodes in yellow are the most recent recurrent keywords, including Backpacking, Tourism Economics, and Economic growth emerging as the keywords concerning the most discussed topics. The violet nodes are the oldest recurrent keywords, including Air Travel, Aged Population, Tourism, Crisis, Globalization, and Access to medicines emerging. The green nodes are from the midpoint of the timeframe of the study, between July and November 2020, and include tourist destination, tourism economics, epidemic, China, and economic growth. Largely, the social science literature has been focusing on the economic aspects of COVID-19 and vaccinations, with a shift in recent months towards the tourism industry.

### 4.1.3. Top Journal and Most Influential Papers

Table 1 reports all journals that published more than one paper dedicated to the social implications from the COVID-19 vaccination, while Table 2 reports the 10 most influential papers by CPM.

**Table 1.** List of sources with more than two publications on the social aspects concerning COVID-19 Vaccinations.

| Analysis of Sources | |
|---|---|
| Journal | Count |
| Health Communication | 2 |
| Ethics and Human Research | 2 |
| R and D Management | 2 |
| Trimestre Economico | 2 |
| Health Communication | 2 |
| European Journal of Risk Regulation | 2 |

**Table 2.** Top 10 most influential papers by number of citations (CPM).

| Top 10 Most Influencial Articles | | | | | |
|---|---|---|---|---|---|
| Authors | Year of Publication | Title | Source (Journal) | Nr. of Cit. | CPM |
| Gössling S., Scott D., Hall C.M. [43] | 2020 | Pandemics, tourism and global change: a rapid assessment of COVID-19 | Journal of Sustainable Tourism | 338.00 | 28.17 |
| Woodside A.G. [44] | 2020 | Interventions as experiments: Connecting the dots in forecasting and overcoming pandemics, global warming, corruption, civil rights violations, misogyny, income inequality, and guns | Journal of Business Research | 10.00 | 1.43 |
| Adongo C.A., Amenumey E.K., Kumi-Kyereme A., Dubé E. [45] | 2021 | Beyond fragmentary: A proposed measure for travel vaccination concerns | Tourism Management | 1.00 | 1.00 |
| Eyal N. [46] | 2020 | Why Challenge Trials of SARS-CoV-2 Vaccines Could Be Ethical Despite Risk of Severe Adverse Events | Ethics and Human Research | 10.00 | 0.91 |
| Fazal T.M. [47] | 2020 | Health Diplomacy in Pandemical Times | International Organization | 6.00 | 0.86 |
| Head K.J., Kasting M.L., Sturm L.A., Hartsock J.A., Zimet G.D. [48] | 2020 | A National Survey Assessing SARS-CoV-2 Vaccination Intentions: Implications for Future Public Health Communication Efforts | Science Communication | 5.00 | 0.71 |
| Fana M., Torrejón Pérez S., Fernández-Macías E. [49] | 2020 | Employment impact of Covid-19 crisis: from short term effects to long terms prospects | Journal of Industrial and Business Economics | 6.00 | 0.67 |
| Banik A., Nag T., Chowdhury S.R., Chatterjee R. [50] | 2021 | Why Do COVID-19 Fatality Rates Differ across Countries? An Explorative Cross-country Study Based on Select Indicators | Global Business Review | 7.00 | 0.64 |
| Chou W.-Y.S., Budenz A. [51] | 2020 | Considering Emotion in COVID-19 Vaccine Communication: Addressing Vaccine Hesitancy and Fostering Vaccine Confidence | Health Communication | 3.00 | 0.50 |
| Farrell R., Michie M., Pope R. [52] | 2020 | Pregnant Women in Trials of Covid-19: A Critical Time to Consider Ethical Frameworks of Inclusion in Clinical Trials | Ethics and Human Research | 4.00 | 0.40 |

We find a diversity of journals that have published manuscripts related to COVID-19 and vaccinations. Yet, there is no clear dominant outlet for these articles. Only 10 journals have more than one publication, indicating the wide-ranging scholarly interest and application of social science approaches to analyzing COVID-19 and vaccinations. The most highly cited articles, even within this short time period of study, analyze the effect of vaccinations on the tourism industry, confirming the analysis conducted of keywords. A second topic emerging from the most cited publications are two papers concerning vaccination intentions of the public, as well as the role of communications from public institutions. An additional two papers analyze the health dynamics for particular subpopulations of patients accessing vaccinations. The remaining cited articles focus on inequalities

in vaccine administration, the ethics of the vaccination process, economic growth spurred from a successful vaccination campaign, and the need for health diplomacy to manage international vaccination campaigns.

### 4.1.4. Most Occurring Countries

Figure 4 looks at the geographic breakdown and LS between countries of articles published on COVID-19 and vaccinations. Similar to above, here the LS indicates the sum cited references that are shared between two countries; the closer two countries represented in Figure 4 indicate an overlapping bibliography and a shared topic of interest. The United States has produced the most papers on the topic of COVID-19 and vaccinations, followed by the United Kingdom and Italy. Indeed, more than 80% of publications prior to May 2020 were published by American researchers. Manuscripts published by researchers in the United Kingdom have been the most influential, with the highest LS, 1035, followed by Canada (620) and New Zealand, Norway and Sweden (all third with an LS of 495). So, while we find evidence that the United States has been at the forefront of producing a high volume of manuscripts on the topic, research produced from the United Kingdom has been more closely linked with research in other countries. Additionally, we find divergence in particular areas of interest by countries. Research produced by the United States, Italy, and Bulgaria focus more on the economic impact of this pandemic, with a particular focus on tourism, while the UK and India concentrate more on the equity of access to care and to the vaccines.

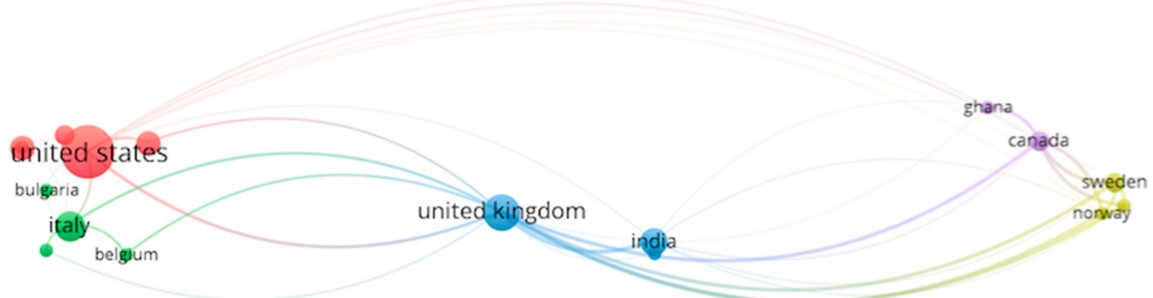

**Figure 4.** Bibliographical mapping of countries.

### *4.2. Bibliometric Analysis*

#### 4.2.1. Cluster Analysis Based on Visualization of Similarities Technique

As described in the methods section, 53 articles were selected for the bibliographic coupling. Out of those 53 articles, only 21 resulted as linked together based on their bibliography (see Table 3). The highest link strength is equal to 8, attributed to Adongo et al. (2021), meaning that the bibliography of this paper shares 8 citations with the overall lists of selected papers. The mean link strength in the list of selected papers is 3.18, while the median is 2.00, meaning that 50% of papers have more than 2 citations in common with other papers within the list. On the other hand, it is possible to appreciate that the average number of citation of papers concerning this topic is lower than the average, if the mean of total citations of the journals with selected papers, according to Scimago (2021), is 42.13, in these selected works is 15.17. As such, an average LS of 2 can be considered a high LS if percentage is taken into consideration (13.18% of common bibliography on average). Figure 5 represents the VosViewer® output and sketches the inter-relation elapsing between the paper bibliographies. Table 3 further specifies this classification, dividing each article into a different cluster identified by VosViewer®. The software identifies five different areas, each one represented by a different color (green, red, violet, blue and yellow). This classification is based only on the bibliographic coupling utilized with the visualization of similarities technique and does not take into consideration the issues addressed in the paper.

**Table 3.** LS for each paper and cluster identified by VosViewer®.

| Authors | Month of Publication | CPM | LS | Cluster |
|---|---|---|---|---|
| De Guttry (2020) [53] | July 2020 | | 5 | |
| Jasso-Villazul and Torres-Vargas (2021) [54] | December 2020 | | 2 | |
| Rizzi (2020) [55] | April 2020 | | 1 | Blue Cluster |
| Phadke (2021) [56] | January 2021 | | 1 | |
| Eyal (2020) [46] | May 2020 | 0.91 | 4 | |
| Gossling et al. (2020) [43] | April 2020 | 28.17 | 4 | |
| Jain and Singh (2020) [57] | December 2020 | 0.25 | 3 | Green Cluster |
| Yotzov et al. (2020) [58] | October 2020 | 0.17 | 1 | |
| Adongo et al. (2021) [45] | March 2021 | 1.00 | 8 | |
| Chou and Budenz (2020) [51] | October 2020 | 0.50 | 6 | |
| Head et al. (2020) [48] | September 2020 | 0.71 | 3 | |
| Gurgula (2020) [59] | October 2020 | | 1 | Red Cluster |
| Guderian et al. (2021) [60] | December 2020 | | 1 | |
| Sell (2020) [61] | November 2020 | | 1 | |
| Arrais et al. (2021) [62] | February 2021 | | 7 | |
| Barbieri Góes and Gallo (2021) [63] | February 2021 | | 7 | Violet Cluster |
| Santoro and Shanklin (2020) [64] | November 2020 | | 1 | |
| Capiku et al. (2021) [65] | September 2021 | 0.14 | 7 | |
| Ma and Ma (2021) [66] | February 2021 | | 2 | |
| Wong (2020) [67] | August 2020 | 0.13 | 1 | Yellow Cluster |
| Fazal (2020) [47] | September 2020 | 0.86 | 1 | |

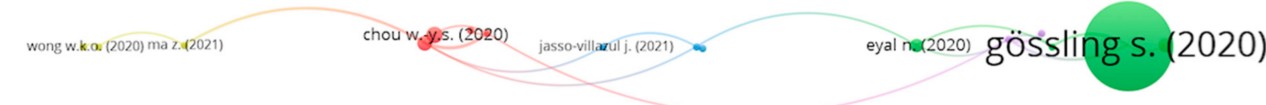

**Figure 5.** Paper clustered by Visualization of Similarities technique identified by VosViewer®.

### 4.2.2. Research Areas (RAs) Identification

To provide a classification that also takes into consideration the content addressed in the articles, we selected papers through the bibliometric clustering process and have been fully and carefully read by different researchers. This process allowed us to identify five main research areas (RAs), representing the five main issues that are being addressed by the current literature. Table 4 represents the RAs identified.

**Table 4.** Articles classification.

| Identified Research Area | Papers |
|---|---|
| RA1—Economic aspects | [63] Barbieri Góes, M. C., & Gallo, E. (2021). Infection Is the Cycle: Unemployment, Output and Economic Policies in the COVID-19 Pandemic. *Review of Political Economy*, 1–17. <br> Jain, V., & Singh, L. (2020). Global spread and socio-economic determinants of Covid-19 pandemic. *Seoul Journal of Economics*, *33*(4). <br> [58] Yotzov, V., Bobeva, D., Loukanova, P., & Nestorov, N. (2020). Macroeconomic Implications of the Fight against COVID-19: First Estimates, Forecasts, and Conclusions. *Economic Studies journal*, (3), 3–28. <br> [61] Sell, S. K. (2020). What COVID-19 Reveals About Twenty-First Century Capitalism: Adversity and Opportunity. *Development*, 1–7. <br> [43] Gössling, S., Scott, D., & Hall, C. M. (2020). Pandemics, tourism and global change: a rapid assessment of COVID-19. *Journal of Sustainable Tourism*, *29*(1), 1–20. |

Table 4. *Cont.*

| Identified Research Area | Papers |
|---|---|
| RA2—Ethics and legal aspects | [53] de Guttry, A. (2020). Is the International Community Ready for the Next Pandemic Wave? A Legal Analysis of the Preparedness Rules Codified in Universal Instruments and of their Impact in the Light of the COVID-19 Experience. *Global Jurist*, *1*(ahead-of-print).<br>[46] Eyal, N. (2020). Why Challenge Trials of SARS-CoV-2 Vaccines Could Be Ethical Despite Risk of Severe Adverse Events. *Ethics & human research*, *42*(4), 24–34.<br>[59] Gurgula, O. (2020). Strategic Patenting by Pharmaceutical Companies–Should Competition Law Intervene?. *IIC-International Review of Intellectual Property and Competition Law*, *51*(9), 1062–1085.<br>[64] Santoro, M., & Shanklin, R. (2020). Human rights obligations of drug companies. *Journal of Human Rights*, *19*(5), 557–567. |
| RA3—Health communication | [45] Adongo, C. A., Amenumey, E. K., Kumi-Kyereme, A., & Dubé, E. (2021). Beyond fragmentary: A proposed measure for travel vaccination concerns. *Tourism management*, *83*, 104180.<br>[51] Chou, W. Y. S., & Budenz, A. (2020). Considering Emotion in COVID-19 vaccine communication: addressing vaccine hesitancy and fostering vaccine confidence. *Health communication*, *35*(14), 1718–1722.<br>[48] Head, K. J., Kasting, M. L., Sturm, L. A., Hartsock, J. A., & Zimet, G. D. (2020). A National Survey Assessing SARS-CoV-2 Vaccination Intentions: Implications for Future Public Health Communication Efforts. *Science Communication*, *42*(5), 698–723.<br>[66] Ma, Z., & Ma, R. (2021). Predicting intentions to vaccinate against COVID-19 and seasonal flu: The role of consideration of future and immediate consequences. *Health Communication*, 1–10. |
| RA4—Policies and Crisis management | [65] Cepiku, D., Giordano, F., Bovaird, T., & Loeffler, E. (2021). New development: Managing the Covid-19 pandemic—from a hospital-centred model of care to a community co-production approach. *Public Money & Management*, *41*(1), 77–80.<br>[54] Villazul, J. J., & Vargas, A. T. (2020). Nuevos mecanismos de colaboración público-privada para el desarrollo y acceso a la vacuna COVID-19: una perspectiva desde la teoría fundamentada. *Contaduría y administración*, *65*(5), 18.<br>[60] Guderian, C. C., Bican, P. M., Riar, F. J., & Chattopadhyay, S. (2021). Innovation management in crisis: patent analytics as a response to the COVID-19 pandemic. *R&D Management*, *51*(2), 223–239.<br>[56] Phadke A. (2021) Questionable Permission, Ineffective Programme: The two recent major decisions of the Indian health authorities regarding Covid-19 vaccination are problematic. *Economic and Poltical Weekly* |
| RA5—Political issues | [62] Arrais, C. A., Corcioli, G., & Medina, G. D. S. (2021). The role played by public universities in mitigating the coronavirus catastrophe in Brazil: solidarity, research and support to local governments facing the health crisis. *Frontiers in Sociology*, *6*, 25.<br>[55] Rizzi, M. (2020). Rethinking vaccine development as integral part of preparednessin the European Health Union. *European Journal of Risk Regulation*, *11*(4).<br>[67] Wong, W. K. O. (2020). China's AI Strike Force on COVID-19. *Asian Education and Development Studies*.<br>[47] Fazal, T. M. (2020). Health diplomacy in pandemical times. *International Organization*, 1–20. |

### 4.2.3. RA1—Economics Aspects

Almost all papers included in this area are focused on macro-economic aspects and the negative impact that the pandemic has had on national economies. One of the main economic aspects emerging from the analysis of papers belonging to this area is on the unemployment rate. Barbieri Góes and Gallo [63] studied the relationship elapsing between non-pharmaceutical actions adopted by governments and the rising unemployment rate. With a two-equations dynamic system model, it is shown that the unemployment rate is subject to fluctuation in the pandemic curve. However, the vaccination represents an important turning point that may invert the general distrust of people (and consequently of firms) across the global economy. The development of vaccinations thus may represent the cure not only for the COVID pandemic but also for the global economy. Additionally, the psychological dimension has to be taken into consideration, as suggested by Yotzov et al. [58], which forecasts the macro-economic effect of the pandemic on the

medium-short run. This means that, under an economic point of view, it is more important to make people believe that vaccines are working rather than having them really working and thus spur economic growth.

However, despite vaccinations being seen as the turning point for the economic fluctuations that characterized the pandemic, it may bring a certain element of moral hazard in the actors developing effective drugs [61]. Therefore, according to Sell [61], the capitalistic push may create a discrepancy in the equal access to care, leading this phenomenon to generate risks and disparities for the global economy.

Another issue concerning the economic influence that vaccinations may have consists of the economic impact of the pandemic in different sectors, with a specific focus on tourism. According to Gössling et al. [43], the concept of tourism and travelling in the forthcoming years will need to be completely reviewed and revamped. Vaccination is an important tool to be used, but does not have to define a discriminatory variable. Rather, it should be exploited to reinforce the economy of those countries that have a large portion of their GDP determined by tourism. If well managed globally, vaccines may represent a valid passport that would allow tourists to be able to travel once again. With this in mind, it cannot represent the panacea for all problems related to the pandemic threat and its impact on tourism. In particular, the psychological dimension behind mass tourism is an important component, when people will be able to be confident once again on being within a crowd visiting touristic spots.

Therefore, if in the first paper [63] we can appreciate the positive aspects of mass vaccination in the other papers included in this area [43,58,61], it is possible to see doubts on their efficacy, not only under a health point of view, but mostly on their influence on economies.

### 4.2.4. RA2—Ethics and Legal Aspects

Certainly, one of the most important areas to be investigated about vaccines and their effect in our society concerns the ethical domain. Strictly related to the ethical issue stands the legal aspect connected to people's vaccination. Papers in this research area raise issues such as the ethical acceptance of trials in order to rapidly develop a vaccine, the competition among firms in the development of a vaccine, possible discrimination within the vaccine campaigns, and other aspects that need further investigation.

The development of a novel vaccine needs to be tested and as such, trials are unavoidable. Furthermore, the will of speeding up the regulatory process governing vaccine development by governments may negatively influence the final outcome.

Indeed, Eyal [46] raises the question of the ethical issue compared to the will of solving the ongoing economic, social, and health crisis. This crisis, according to the author, "cannot be sustainably resolved without a proven vaccine against the novel Coronavirus". The main point here is whether speeding up the process of phase III (notably, the longest phase of vaccination testing), and substantially increasing the risk for human life, can be considered morally and ethically appropriate in such circumstances.

Related to the issue of speeding up the vaccine process, another issue that needs to be considered is the competition between pharmaceutical companies. This challenge is summarized by the patenting issue, as pointed out by Gurgula [59]. Strategic patenting, which is considered a legal practice under certain political systems, may threaten the distribution of vaccines and compromise the whole success of the campaign. More specifically, Gurgula [59] argues the effect of strategic patenting actions performed by pharmaceutical companies, and criticizes its use in situations of a general threat to the whole world. However it gives a critical perspective of the issue, suggesting that governments should directly intervene to regulate and administer the role of private actors. Strategic patenting involves prosecuting patent applications with the sole purpose of improving a firm's patent portfolio in a specific technological area. Due to the dimension of this issue, if applied to the COVID-19 pandemic, the moral question appears clear. Gurgula [59] concludes his analysis with a specific warning: "while currently strategic patenting is considered a

lawful practice, it should attract the attention ( . . . ) national competition authorities" as this concept in this specific era may represent a threat to society in the name of profit.

This conclusion is also supported by [64], as they deal with human rights obligations of drug companies in relation to the spread of the pandemic and the possible panacea of mass vaccination. Specifically, their work "addresses the human obligations of the pharmaceutical industry with regard to vaccines and treatments developed to prevent and treat Covid-19", enlightening the concept of corporate social responsibility and identifying few tools that could help in preventing opportunism and exploitation by private pharmaceutical companies, such as the use of corporate philanthropy as well as government policies apt to incentivize the fair share.

Finally, de Guttery [53] analyzes the legal impact of an international distribution of COVID-19 impact highlighting possible issues arising from different systems forced to cooperate in such a situation.

### 4.2.5. RA3—Health Communication

Health communication is particularly important during the pandemic, as information and social media spread quickly, and too often so does inaccurate science. Anti-vax movements already represent an important problem and a big issue faced by governments. The advent of the COVID-19 pandemic amplified this problem and, as such, an effective communication by health authorities has been and still is crucial.

Another important problem addressed by previous literature consists of the travel vaccination concerns, as often the importance of vaccination while travelling, according to Adongo et al. [45], is not sufficiently highlighted by health authorities. Therefore, this area, which includes four different published papers, addresses these two main concerns, yet few conclusions are made.

More specifically, Adongo et al. [45] addresses the issue arising from the travel vaccination in the time of COVID-19 pandemics. This is actually not a new concept, as it was already widely discussed by previous scholars. One of the key points made in the introduction of this article entails affirming that this practice highlights inequalities between citizens and nations. The purpose of this study is therefore attempting to sketch and tailor strategies in order to avoid this issue. The study itself consists of a SLR combined with a quantitative analysis of a survey-questionnaire administered to vaccine travellers. The general idea is preventing tourists to travel without proper vaccinations, whether made in their own countries or upon their arrivals. The authors conclude that health communication that addresses vaccination concerns needs deeper consideration by authorities, and better tailored health communication by governments may be the most effective tool to prevent this type of opportunistic behaviour. Therefore, implementing and investing in campaigns apt to promote the idea of vaccinating tourists upon their arrivals, if not already vaccinated in their country, combined with pricing policies may be the key to re-open up to tourism.

Chou and Bundez [51], instead, are more focused on the vaccination concerns among individuals and they investigate the role of emotions over the choice of being or not being vaccinated. Misinformation, combined with the radical social change imposed by the pandemic, may result in a population having trouble managing emotions. The duty of administrations is to do whatever they can in order to prevent this. The authors here propose a few solutions: (1) concentrating all vaccination efforts in one community and then completely re-open up that same community in order to positively influence public opinion towards COVID-19 vaccination; (2) fostering individuals' self-efficacy on vaccination may generate positive emotions and help people in gaining confidence with the drug; (3) providing public space for information campaigns in order to contrast disinformation and lead people towards the generation of positive emotions.

Another point on health communication was made by the work of Head et al. [48], which investigated the intentions of people of being vaccinated before the Sars-CoV-2 vaccines were placed on the market. Their analysis focuses on the social variables that may

influence vaccination intentions. Their conclusions highlight the importance of education and the role that governments need to have in order to encourage responsible behaviour.

Finally, Ma and Ma [66] attempted to forecast intentions of vaccinating based on seasonal flu vaccination data and previous surveys. The purpose of this study was to sketch the situation at the time it was written and help governments in providing a suitable way to measure vaccination intentions and possibly tailoring effective strategies to cope with vaccine hesitancy or antagonism. However, it turned out that their predictions, which proposed a substantially negative situation, did not match what has been the reality, suggesting that overall western governments addressed effective campaigns to fight misinformation.

### 4.2.6. RA4—Policies and Crisis Management

RA4 focuses on crisis management literature and policies undertaken to cope with crisis. This is not a new issue for health policy scholars and policy-makers, as historical events have shown how crises cannot be ignored, despite being rare, they can bring long-term and dramatic consequences [68]. Recent, as well as older literatures have provided insights on how it can be dangerous to not build a resilient model to react to crisis when they occur [69,70].

In this research area, issues are addressed related to the management of vaccinations and the policy created in order to maximize its efficacy. Villazul and Vargas [54] elaborated a perspective in which they analyze and explain why public–private partnership in this context may be the best solution. They reach the conclusion that a long-term cooperation in this sector, which should not be only a public prerogative, makes ideally possible a more effective impact of the vaccinations. Indeed, if public and private entities collaborate sharing the resources, risks and costs related to the vaccination campaign, both will engage in a fruitful relationship.

Phadeke [56] questions policies adopted but some of the largest countries in the world in terms of population (focusing primarily on India) argue that it is necessary to design a resiliency model before a crisis happens, providing flexibility and alongside a clear path to follow. In countries where this did not occur, the authors argue that problems containing and addressing COVID were most evident.

Finally, Cepiku et al. [65] did not perform an analysis specifically based on vaccinations, but their conclusions, where they propose an organizational model of health (which also include vaccination campaigns) more centered on communities rather than hospitals. In other words, a model providing more attention to the territorial medicine and its effects on ameliorating the effects of COVID-19.

In summary, all these articles criticize the absence of a structured model suggesting that these experiences should be exploited to make resilient plans in order to better cope with future crises.

### 4.2.7. RA 5—Political Issues

Historically, health has lingered on the sidelines of international relations [47]. The politics of health emerging from COVID-19 and the importance of international politics thus indicates a potential shifting in these views. Therefore, issues faced in this RA concern some of the most important political problems concerning vaccination.

More specifically, Rizzi [55] and Fazal [47] investigate the influence that vaccinations may have on international relationships between countries. They first argue that the EU should rethink its vaccine policy and better coordinate between all country members in order to provide a more efficient plan and bring a sense of integration to European people, whilst Fazal [47] investigates the role of health diplomacy during the whole COVID-19 crisis, with a specific focus on vaccination.

Arrais et al. [62] performed an analysis of what has been the main role of the public university in fighting the pandemic and also organizing the vaccine campaigns in

South America, whilst Wong [67] provided a study showing problematics concerning the management of the crisis in China.

All the papers reach a simple but important conclusion: vaccination is a matter of politics and as such should be addressed by scholars;this perspective is still under-represented in the literature.

## 5. Conclusions

COVID-19 has upended life around the globe. One of the most important developments in the effort to contain and mitigate the pandemic has been the development of vaccines. While social science researchers have developed a rich set of literatures around vaccination, the emergence of COVID-19 requires an evaluation of what we have learned about the pandemic. Here, we have completed the first systematic literature review of social science research on COVID-19 and vaccinations.

There are two themes which have emerged from our analysis. First, the development of research on the topic has not been uniform across all topics. We find that much of the focus of published literature has focused on the effects of COVID-19 and vaccinations through an economic perspective. This has been manifested largely in the focus on economic growth and tourism. This indicates to us that there are other areas of social science which are ripe for exploration and understanding the effects of the development of vaccines on COVID-19.

Second, our analysis raises some concerns about the global application of the already completed work on vaccines and COVID-19. Not surprisingly, we find that research conducted in countries such as the United States and United Kingdom dominate published work across the globe. Yet, we also find a startingly small number of works in countries representing Asia, Africa, and South America. As issues with the equitable distribution of vaccines across the globe and the threat of COVID-19 is projected to continue in low- and middle-income countries, research is urgently needed to focus on those areas of the world. This issue is starting to be partially covered at this time, for instance the valuable work of Bolcato et al. [71] has brought the attention of equal access to vaccination all over the world and what barriers may prejudice its development. Alongside Bolcato, a few other works have investigated the same issue, including Aborode et al. [72], Jecker et al. [73], and Gastrow and Lawrence [74].

Research on vaccine hesitancy, distribution challenges, political arrangements, and other topics which are situated in those regions and countries will be necessary because the research findings that have already been developed are not "one-size-fits-all".

Taken together, the results of our systematic review paint a decidedly mixed picture. Social science researchers have quickly been able to produce a sizable amount of research in very short order. This is to be commended, particularly in a field where the norm has traditionally been slow to respond to emergent issues. COVID-19 has had such wide-ranging effects that researchers have found a variety of different angles, and thus have been able to pursue their work in interesting and unique ways. Yet, the results also paint a picture of literature that has not yet considered the full scope of COVID-19s effect on the economic, political, and population level health and well-being. Nor has it considered these effects across the global community. Future work, both by researchers and especially journal editors, should focus on a broader spectrum of understanding how COVID-19 and vaccinations affect society.

**Author Contributions:** Introduction, L.P. and P.M.S.; Literature background, L.P. and P.M.S.; Methodology, L.P. and P.M.S.; results, L.P. and P.M.S.; Conclusions, L.P. and P.M.S. All authors have read and agreed to the published version of the manuscript.

**Funding:** This research received no external funding.

**Institutional Review Board Statement:** Not applicable.

**Informed Consent Statement:** Not applicable.

**Data Availability Statement:** Data sharing not applicable.

**Conflicts of Interest:** The authors declare no conflict of interest.

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
