# Peer review of "COVID-19 Vaccination: What Do We Expect for the Future? A Systematic Literature Review of Social Science Publications in the First Year of the Pandemic (2020–2021)"

_sustainability, doi:10.3390/su13158259_

Round 1

Reviewer 1 Report

This systematic literature review analyzed social science publications related to COVID-19 and vaccination in the first year of the pandemic (February 2020 to March 2021). I think the research is timely and relevant, and provides valuable directions for future research. However, I have some questions and concerns about the data collection and the scope of this review. I elaborate on my comments below.

  1. I suggest the authors adding a timeframe to the title. This study only reviewed research published in the first year of the pandemic 15 (February 2020 to March 2021). As more studies on COVID-19 vaccine emerge, it is important to specify the timeframe of this review. I’d suggest the authors adding “2020-2021” at the end of the title to be more specific and accurate.

  1. SARS-CoV-2 is the virus, and COVID-19 is the disease caused by SARS-CoV-2. To be more accurate, I suggest the authors using “The COVID-19 pandemic” in the first sentences in the Abstract and the Introduction.

  1. Lines 71-73: I don’t understand this sentence. Do the authors try to argue that vaccines are as important as clean water? I’d suggest removing this sentence.

  1. The authors mentioned, “a clear strand focused on COVID-19 vaccination is social sciences has yet to be built” (Lines 114-115). While I understand that few studies have explicitly focused on COVID-19 vaccination during the timeframe of this review, I am wondering how appropriate and valid it is to include and review studies that just briefly mentioned COVID vaccine/vaccination. That is, “no clear strand” does not automatically justify that we should include papers that focused on “general issues caused by the pandemic” in a COVID vaccination literature review. For instance, the most cited study, Gossling et al. 2020, had no direct focus on COVID vaccinations. The few times when the article mentioned COVID vaccine, it stated that “in the absence of the vaccine.” I am not sure how and why this is considered a COVID vaccination publication. Thus, I strongly suggest the authors (a) clearly defining what they mean by “COVID vaccination publication,” (b) clearly defining their inclusion criteria (see Comment 7 below), and (c) more convincingly justify their inclusion criteria.

  1. Figure 1, the number in the third red arrow is missing.

  1. Lines 168-169: Please specify that the “keywords” included words in the title, words in the “key words” section of the article, and words in the text. Take Gossling et al. 2020 as an example again, the title and key words of the article did not have “vaccine” or “vaccination.” “Vaccine/ vaccination” only appeared in the text.

  1. What are the criteria to finalize the 53 articles in the review? How did the authors determine if an article was relevant or not?

  1. Lines 176-182: Please provide more information about the intercoder reliability. Did both coders code all 121 articles? How did the authors/research team solve the coding discrepancies?

  1. I don’t understand Table 1. Each journal appeared twice, and all the counts were 2. Is the table informative?

  1. Assuming the data collection was valid, I think the analysis and results were valid, appropriate, and meaningful. I’d suggest the authors attending to the data collection and inclusion criterion issues I pointed above.

Author Response

Comments and Responses to Reviewer #1

Reviewer:

This systematic literature review analyzed social science publications related to COVID-19 and vaccination in the first year of the pandemic (February 2020 to March 2021). I think the research is timely and relevant, and provides valuable directions for future research. However, I have some questions and concerns about the data collection and the scope of this review. I elaborate on my comments below.

Authors:

Thank you for the praising comment. We will address your concerns in the following lines.

Reviewer:

I suggest the authors adding a timeframe to the title. This study only reviewed research published in the first year of the pandemic 15 (February 2020 to March 2021). As more studies on COVID-19 vaccine emerge, it is important to specify the timeframe of this review. I’d suggest the authors adding “2020-2021” at the end of the title to be more specific and accurate.

Authors:

Thank you for the precious comment. We indeed updated the title.

Reviewer:

SARS-CoV-2 is the virus, and COVID-19 is the disease caused by SARS-CoV-2. To be more accurate, I suggest the authors using “The COVID-19 pandemic” in the first sentences in the Abstract and the Introduction.

Authors:

We substituted Sars-CoV-2 with Covid-19 when needed. However, we also specified that Sars-CoV-2 is the virus that caused the COVID-19 pandemic. The objective of this was to make it clearer for the reader (even if probably it is not necessary at the present time), since some of the publications we analyse used the “Sars-CoV-2” expression.

Reviewer:

Lines 71-73: I don’t understand this sentence. Do the authors try to argue that vaccines are as important as clean water? I’d suggest removing this sentence.

Authors:

Thank you for the comment and clarification. Our intent was not to compare vaccines with clean water, merely to highlight the importance of vaccinations. We agree with the reviewer suggestion and have deleted the sentence.

Reviewer:

The authors mentioned, “a clear strand focused on COVID-19 vaccination is social sciences has yet to be built” (Lines 114-115). While I understand that few studies have explicitly focused on COVID-19 vaccination during the timeframe of this review, I am wondering how appropriate and valid it is to include and review studies that just briefly mentioned COVID vaccine/vaccination. That is, “no clear strand” does not automatically justify that we should include papers that focused on “general issues caused by the pandemic” in a COVID vaccination literature review. For instance, the most cited study, Gossling et al. 2020, had no direct focus on COVID vaccinations. The few times when the article mentioned COVID vaccine, it stated that “in the absence of the vaccine.” I am not sure how and why this is considered a COVID vaccination publication. Thus, I strongly suggest the authors (a) clearly defining what they mean by “COVID vaccination publication,” (b) clearly defining their inclusion criteria (see Comment 7 below), and (c) more convincingly justify their inclusion criteria.

Authors:

Thank you for this precious comment. In fact, this has been the most critical point of the analysis and has been discussed for long time by the authors. We wanted to have a consistent amount of paper to perform a literature review, but basing our choice only on titles and abstract made not possible to achieve this outcome. Therefore, we went through all the papers and we decided to include also papers that had relevant section concerning vaccinations.

More precisely, the mentioned paper of Groessling et al. has a specific paragraph in which it is enlightened what will be the effect of vaccines on tourism. The paper doesn’t merely say that vaccination is the only solution, but defines possible timelines stating that it will be necessary a period of time between 12 and 18 months to have it ready and at least the same time to return back to normal, with the implications that this have on firms working in the tourism industry. It is true that it does not refer only to vaccination, but the same authors state highlight the importance of it in order to make travelling safe.

We report the paragraph that made us convinced of choosing this article as relevant:

Once a vaccine is developed and received authorization for use, phase three (establish immune protection and life physical distancing) physical distancing restrictions and other NPIs can be lifted. Once phase three and widespread vaccination is completed, global tourism will be safe to recommence. Tremendous research is being done to fast-track the development and testing of vaccines, but the estimated timeline remains 12-18 months. The final phase (rebuild readiness for next pandemic) needs to invest in research and disease monitoring, health care infrastructure and workforce, and improve governance and communication structures. Tourism, in particular air travel and airports, must be part of new international monitoring and rapid response plans. This would also include a better understanding of tourism’s role in pandemics: Air travel and transport more generally support the spread of pathogens, while the sector also contributes to growing pressure on remaining forest ecosystems (through land use or industrial food sourcing), i.e. developments that are seen to increase the likelihood of future pandemics”.

We then better specify what we meant on page 6, line 16. 

Reviewer:

Figure 1, the number in the third red arrow is missing.

Authors:

Thank you. We added it. 

Reviewer:

Lines 168-169: Please specify that the “keywords” included words in the title, words in the “key words” section of the article, and words in the text. Take Gossling et al. 2020 as an example again, the title and key words of the article did not have “vaccine” or “vaccination.” “Vaccine/ vaccination” only appeared in the text.

Authors:

Thank you. We added it. 

Reviewer:

What are the criteria to finalize the 53 articles in the review? How did the authors determine if an article was relevant or not?

Lines 176-182: Please provide more information about the intercoder reliability. Did both coders code all 121 articles? How did the authors/research team solve the coding discrepancies?

Authors:

Thank you for the comment. In fact, we did not specify that further to not make the methodology section to heavy to read. However, at page 8 line 14 we added the explanation on how we have ended up with 53 articles, how we determined the relevancy of the articles and how we reached the ICR (Inter-coder reliability). We report the explanation here:

To determine the ICR “we followed the methodology designed by the same Cohen in 1960 as well as by Zade et al. (2018). The two researchers-judges had to use a chromatic semantic-dimensional scale 1 to 5 to indicate how relevant was the article (dark red = 1, non-relevant; orange = 2, very low relevancy; yellow = 3, possible relevancy; light green = 4, relvant; dark green = 5, very relvant). The judgement has been double blind and after, in the revision phase, when a discrepancy of more than 2 points was found, a discussion between the judges and all the authors of the paper was conducted. Finally, the Cohen’s Kappa was calculated to determine the inter-reliability of the choices and with a 0.8794 Cohen’s K, this score has been considered acceptable.”

Reviewer:

I don’t understand Table 1. Each journal appeared twice, and all the counts were 2. Is the table informative?

Authors:

Thank you for the comment. In fact, the fact that each journal appeared twice is a wrongly made process of copy and paste. We adjusted the table. The table has an informative purpose in order to give a perspective of what journals have published more so far on the social science related vaccine issues.

Assuming the data collection was valid, I think the analysis and results were valid, appropriate, and meaningful. I’d suggest the authors attending to the data collection and inclusion criterion issues I pointed above.

Authors:

Thank you. We hope we made our purpose clearer and that we solved the issues pointed above by the reviewer. We also would like to thank the reviewer to have helped us in improving this work.

Reviewer 2 Report

I have read the article with interest and I believe that it is built with a good methodology and an important work by the authors. The research base is interesting and certainly useful for readers. I think the ethics and legal aspects can be at least a little in-depth. In particular, several articles like this DOI: 10.3390 / vaccines9060538 indicate, in addition to the aspects that have been mentioned, the need for fairness in vaccination distribution even in non-industrialized countries. This aspect will be increasingly important for the return to normalcy throughout the planet. I think you could briefly mention this study. For the rest I believe that the article can be published

Author Response

Comments and Responses to Reviewer #2

Reviewer

I have read the article with interest and I believe that it is built with a good methodology and an important work by the authors. The research base is interesting and certainly useful for readers.

Authors:

We would like to thank the reviewer for the praising comments.

Reviewer

I think the ethics and legal aspects can be at least a little in-depth. In particular, several articles like this DOI: 10.3390 / vaccines9060538 indicate, in addition to the aspects that have been mentioned, the need for fairness in vaccination distribution even in non-industrialized countries. This aspect will be increasingly important for the return to normalcy throughout the planet. I think you could briefly mention this study. For the rest I believe that the article can be published

Authors:

Thank you for the suggestion: the paper is relevant and can fit into the analysis. However, it was published in April 2021 while our research refers to all publications between January 2020 and March 2021. However, since the paper is very relevant, we decided to mention it, as well as few other studies that have just begun to address the same issue, in the conclusion section.

Round 2

Reviewer 1 Report

The authors addressed my comments well. I don't have further comments or concerns. Thank you for this important work.